# KptLLM: Unveiling the Power of Large Language Model for Keypoint Comprehension

**Jie Yang**[1,2,5] **Wang Zeng**[4,5] **Sheng Jin**[3,5] **Lumin Xu**[4] **Wentao Liu**[5*] **Chen Qian**[5] **Ruimao Zhang**[1*]

[1] Sun Yat-sen University  [2]The Chinese University of Hong Kong, Shenzhen
[3]The University of Hong Kong  [4]The Chinese University of Hong Kong
[5] SenseTime Research and Tetras.AI

`https://kptllm.github.io`

## Abstract

Recent advancements in Multimodal Large Language Models (MLLMs) have greatly improved their abilities in image understanding. However, these models often struggle with grasping pixel-level semantic details, *e.g.*, the keypoints of an object. To bridge this gap, we introduce the novel challenge of *Semantic Keypoint Comprehension*, which aims to comprehend keypoints across different task scenarios, including keypoint semantic understanding, visual prompt-based keypoint detection, and textual prompt-based keypoint detection. Moreover, we introduce KptLLM, a unified multimodal model that utilizes an identify-then-detect strategy to effectively address these challenges. KptLLM underscores the initial discernment of semantics in keypoints, followed by the precise determination of their positions through a chain-of-thought process. With several carefully designed modules, KptLLM adeptly handles various modality inputs, facilitating the interpretation of both semantic contents and keypoint locations. Our extensive experiments demonstrate KptLLM's superiority in various keypoint detection benchmarks and its unique semantic capabilities in interpreting keypoints.

## 1 Introduction

Recent advancements in deep learning and natural language processing have facilitated the rise of Large Language Models (LLMs) that display human-like fluency in text comprehension and generation [1, 2, 3, 4, 5, 6, 7]. By incorporating visual information, researchers have developed Multimodal Large Language Models (MLLMs) [8, 9, 10, 11, 12], specifically designed for visual-language tasks, showcasing remarkable abilities in image understanding. However, these models encounter difficulties in capturing fine-grained semantic details, particularly at the point level, which are crucial for various real-world applications. The exploration of MLLMs for keypoint comprehension remains under-explored in the literature.

Keypoint detection is a fundamental aspect of computer vision that supports various applications such as controllable image/video generation [13, 14, 15], human-centric perception [16, 17], and AR/VR systems [18, 19, 20]. Initially, research in this field focused on closed-set problems, aiming to predict the locations of predefined semantic keypoints of a certain object category (*e.g.* human body). As the demand for generalization grew, researchers started investigating the detection of keypoints for novel objects by providing visual prompts (*i.e.*, a support image of a novel object with its keypoint definitions) [21, 22] or utilizing textual prompts (*i.e.*, keypoint names) [23]. Despite significant progress in these areas, existing models still fall short of achieving genuine semantic comprehension

---

[*]Corresponding author

38th Conference on Neural Information Processing Systems (NeurIPS 2024).

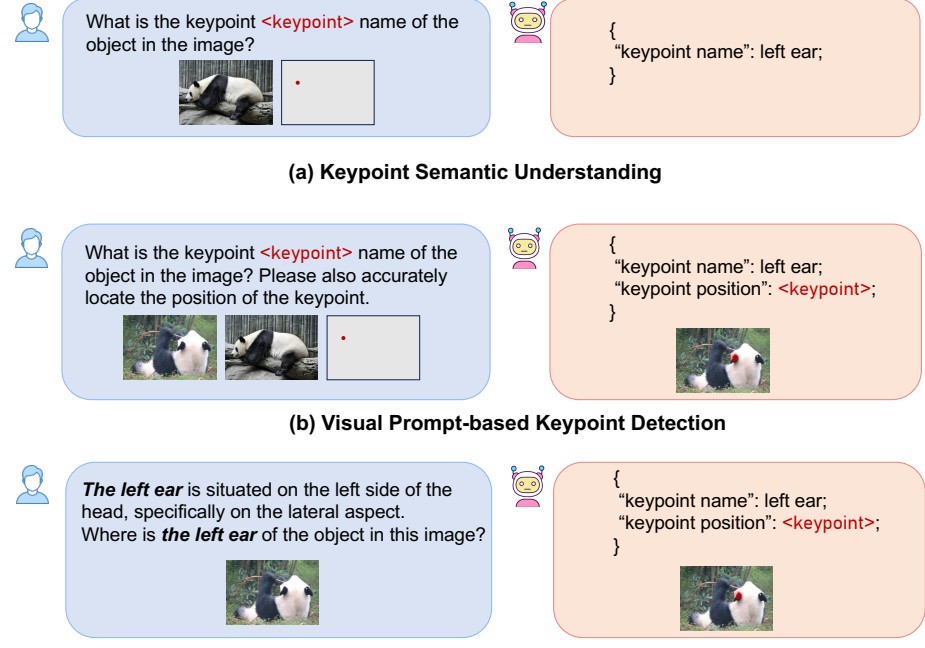

(a) Keypoint Semantic Understanding

(b) Visual Prompt-based Keypoint Detection

(c) Textual Prompt-based Keypoint Detection

Figure 1: This work aims to address the problem of semantic keypoint comprehension, which aims to understand keypoints across different task scenarios: *(a) Keypoint Semantic Understanding* takes the object image and a keypoint prompt (*i.e.*, the position of the target keypoint) as inputs, then generate responses that interpret keypoint semantics; *(b) Visual Prompt-based Keypoint Detection* takes a query image and a support image with a keypoint prompt as inputs and then outputs the corresponding keypoint positions and semantics of the query image; *(c) Textual Prompt-based Keypoint Detection* utilizes detailed descriptions of keypoints through extensive text, to perform more generalizable keypoint detection.

of keypoints akin to humans. These models primarily rely on direct learning of visual patterns for keypoint localization through extensive data fitting, while neglecting semantic understanding of the keypoints, thus leading to misinterpretation of the prompts and inaccurate predictions. Moreover, the input-output structures are designed in fixed and predefined formats, restricting their usage to predetermined methods and impeding the flexibility required for interfacing with users.

Motivated by the aforementioned challenges, this paper delves into a more comprehensive problem of *Semantic Keypoint Comprehension* to evaluate the model capability of comprehensively understanding keypoints both visually and semantically. As shown in Fig. 1, we investigate three distinct capabilities via different task instructions: *(a) Keypoint Semantic Understanding* aims to infer the desired keypoint semantics, given the target image and a keypoint prompt (*i.e.*, the position of the target keypoint) as inputs. It provides the potential for an AI model with high-level visual understanding and analytical capabilities, crucial for tasks such as structural comprehension, action recognition, and medical image analysis. *(b) Visual Prompt-based Keypoint Detection*, also referred to as category-agnostic pose estimation, takes a query image and a labeled support image with the keypoint annotation as inputs and then outputs the corresponding keypoint positions in the query image. This capability requires the model to acquire keypoint definitions from visual prompts, enabling it to perform cross-class and cross-keypoint localization tasks using sample images provided by users. *(c) Textual Prompt-based Keypoint Detection*, also known as open-vocabulary keypoint detection, aims to utilize detailed descriptions of keypoints through extensive text for keypoint localization. The keypoint detectors directly receive the human language guidance, facilitating keypoint localization on arbitrary object and keypoint categories in a zero-shot manner.

We introduce KptLLM, a novel framework that utilizes an identify-then-detect strategy to address the challenging problem of semantic keypoint comprehension. It formulates all three capabilities depicted in Fig. 1, by first identifying the semantic meaning of keypoints and then detecting their positions via a chain-of-thought approach, akin to human cognition. KptLLM is a unified framework

that comprises four key components designed to accommodate various modality inputs and infer both the semantics and location of the keypoint. Specifically, we first extract visual features of both query and support images to obtain query visual tokens and support image features. Secondly, we encode the support keypoint prompt, which describes the position of keypoint on the support image, to generate keypoint prompt embedding. Thirdly, prompt-oriented features are derived by integrating support image features with keypoint prompt embedding, and are utilized to form keypoint prompt tokens. Lastly, LLMs take query visual tokens, keypoint prompt tokens, and task-related language tokens as input, and then generate the semantic description of the target keypoint and its corresponding position on the query image. By harnessing commonsense knowledge in LLMs, KptLLM can assist in keypoint localization of novel object categories, potentially leading to enhanced generalizability in performance. In addition, the chain-of-thought design elicits the powerful keypoint understanding capabilities of LLMs, which helps to distinguish visually ambiguous keypoints (*e.g.* left and right arms). Extensive experiments demonstrate KptLLM's superiority on semantic keypoint comprehension, showcasing its unique semantic understanding capabilities in interpreting keypoints and state-of-the-art performance in various keypoint detection benchmarks.

In summary, the contributions of this work are three-fold: (1) We pioneer the investigation of a novel problem in semantically interpretable keypoint analysis, termed *Semantic Keypoint Comprehension*, which aims to enhance MLLMs with improved image understanding at a finer-grained keypoint level; (2) We introduce KptLLM, a unified multimodal model that utilizes an identify-then-detect strategy to effectively address three tasks of semantic keypoint comprehension. KptLLM underscores the initial discernment of semantic significance in keypoints, followed by the precise determination of their positions through a chain-of-thought process. (3) We demonstrate KptLLM's superiority in various existing keypoint detection benchmarks and its unique semantic capabilities in interpreting keypoints. We hope our work could inspire future research on keypoint understanding and localization, while also fostering enhanced human-AI interface in fine-grained visual understanding.

## 2 Related Work

### 2.1 Keypoint Detection

Keypoint detection, also referred to as pose estimation, focuses on localizing the 2D keypoints of objects in the image. Traditional models for keypoint detection are typically designed for a single category, *e.g.*, human [24, 25, 26, 27, 28, 29, 30], animal [31, 32, 33] and clothes [34]. Based on the localization strategy, existing methods are generally divided into regression-based methods [35, 36, 37, 38, 39, 40] and heatmap-based methods [41, 42, 43, 44, 45, 46, 47, 48, 49, 50, 51]. More recently, methods that can recognize and localize keypoints for unseen object categories in the training datasets, are gaining increasing attention from the community. Category-agnostic pose estimation [21, 22], also referred to few-shot keypoint detection, aims to estimate the pose of any category in query images with visual prompts (*i.e.*, a few support images of a novel class and its corresponding keypoint annotations). Another line of research explores open-vocabulary keypoint detection [23, 52], which aims to localize keypoints based on text prompts in zero-shot settings. In this work, we investigate semantic keypoint comprehension and propose a novel unified framework to comprehend keypoints across three different task scenarios, including (a) keypoint semantic understanding; (b) visual prompt-based keypoint detection; (c) textual prompt-based keypoint detection.

### 2.2 Multimodal Large Language Model

Inspired by the success of Large Language Models (LLMs) [1, 2, 53, 3, 4, 54, 5, 6, 7], researchers are exploring ways to transfer the formidable capabilities of LLMs into the realm of vision, developing Multimodal Large Language Models (MLLMs) [55, 8, 9, 10, 56, 57, 11, 12, 58, 59, 60, 61, 62, 19]. These models exemplify an autoregressive mechanism predicated on a transformer decoder architecture [63]. The integration of visual representation from vision encoders [64, 65] into the domain of LLMs ushers a new era of visual comprehension and reasoning. Such integration is predominantly facilitated through a Multilayer Perceptron (MLP) that seamlessly transforms visual features into the input embedding space of LLMs [10, 56, 57], or via a cross-attention mechanism that attends to visual contents through a series of attention layers [55, 8, 11]. However, most of these VLMs can only provide text outputs, inhibiting the complex applications requiring detailed visual perception. VisionLLM [66] tackles a range of conventional vision-centric tasks by instruction

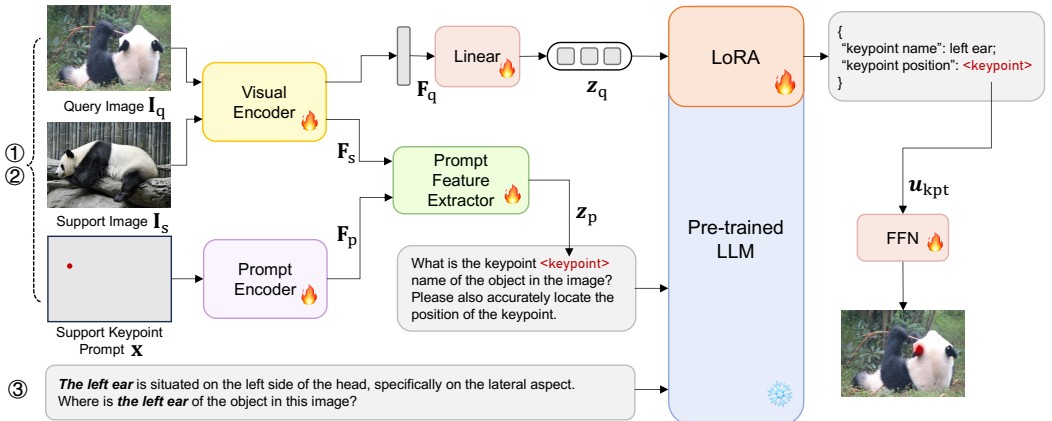

Figure 2: We introduce KptLLM, a unified framework designed to address three tasks of semantic keypoint comprehension: ① ***Keypoint Semantic Understanding***, which processes a support image $\mathbf{I}_s$ and a support keypoint prompt $\mathbf{x}$ to generate responses that interpret the semantic information of the specified keypoint; ② ***Visual Prompt-based Keypoint Detection*** aims to detect the corresponding keypoint in the query image $\mathbf{I}_q$ based on the understanding of the support keypoint prompt; ③ ***Textual Prompt-based Keypoint Detection*** leverages textual keypoint descriptions to directly infer the corresponding keypoint positions in the query image.

tuning LLMs. However, it may fall short of fully leveraging the comprehensive reasoning faculties of LLMs. Kosmos-2 [67], Qwen-VL [68] and DetGPT [69] further exploit the power of LLMs to enable user-guided detection. Moreover, GPT4RoI [70], Ferret [71], Shikra [72], and PerceptionGPT [73] innovates by incorporating spatial boxes or masks as inputs and training with region-text pairs, offering region-level visual comprehension. Notably, a concurrent work, LocLLM [74], utilizes LLMs for human keypoint localization via textual description. In contrast, we take a step further by enabling LLMs to comprehend keypoints of various objects via multi-modal (*e.g.*, textual or visual) prompts under different task formulations. This advancement not only broadens the utility of MLLMs for keypoint detection but also enhances interpretive depth, allowing for a more comprehensive understanding and grounding across a wider range of visual information.

## 3 Methodology

This section introduces our proposed unified framework, referred to as KptLLM, which effectively addresses three semantic keypoint comprehension scenarios. As illustrated in Fig. 2, KptLLM accepts multiple images (query and support images) along with a support keypoint prompt (*i.e.*, the position of the target keypoint in the support image) and textual user instructions as the input. The output comprises both the response text and the desired keypoint position. Specifically, KptLLM comprises four key architectural components: (1) A visual encoder that extracts features from both query and support images (see Sec. 3.1); (2) A prompt encoder that converts support keypoint prompts into prompt embeddings (see Sec. 3.2); (3) A prompt feature extractor that derives prompt-oriented features from the corresponding image features (see Sec. 3.3); (4) A pre-trained LLM that processes multimodal tokens for keypoint comprehension (see Sec. 3.4).

### 3.1 Visual Encoder

The Visual Encoder is designed to process two types of images in parallel: query and support images. Generally, it receives an input image $\mathbf{I} \in \mathbb{R}^{H \times W \times 3}$ and generates a feature map $\mathbf{F} = \mathcal{V}(\mathbf{I}) \in \mathbb{R}^{h \times w \times d}$. Here, $d$ represents the feature dimension, and $h$ and $w$ are the spatial dimensions obtained by downsampling the original image dimensions $H$ and $W$.

**Query Image.** The query image represents the image that is to be analyzed. We extract its spatial features through the vision encoder $\mathcal{V}$, resulting in $\mathbf{F}_q$. Following LLaVA [10], we apply a linear layer to project $\mathbf{F}_q$ into language space: $\mathbf{z}_q = \texttt{Linear}(\mathbf{F}_q)$. As a result, query visual tokens aligned with the LLM dimension are obtained and fed to the LLM.

**Support Image.** The support image serves as a reference example. We extract its spatial features, which are represented as $\mathbf{F}_s$. Unlike the query image features, $\mathbf{F}_s$ is not directly input into LLM. Instead, it is processed by the prompt feature extractor to derive prompt-oriented features.

## 3.2 Prompt Encoder

In addition to processing images, we need to incorporate an additional prompt consisting of 2D coordinates $\mathbf{x} \in \mathbb{R}^2$, which describes the keypoint location within the image. Inspired by SAM [75], we introduce a prompt encoder to adapt this prompt input to be aligned with the image feature space $\mathbf{F}$. The prompt encoder encodes the keypoint coordinates using a sine-cosine position embedding (PE), followed by a Multi-Layer Perceptron (MLP):

$$\mathbf{F}_p = \texttt{MLP}(\texttt{PE}(\mathbf{x})). \tag{1}$$

## 3.3 Prompt Feature Extractor

The Prompt Feature Extractor is designed to extract the prompt-specific features from image features. As illustrated in Fig. 2, the semantics of the keypoint prompt directly correspond to the support image. We initialize the prompt feature extractor with a two-layer transformer that incorporates the cross-attention mechanism (`CrossAttnLayers`). This mechanism employs $\mathbf{F}_p$ as the query and $\mathbf{F}_s$ as the key and value to extract keypoint-specific visual features indicated by the prompt:

$$\mathbf{z}_p = \texttt{CrossAttnLayers}(\mathbf{F}_p, \mathbf{F}_s), \tag{2}$$

where $\mathbf{z}_p$ denotes the keypoint-specific visual features. In essence, compared with average pooling-based feature extraction method [21], the prompt feature extractor is trainable and capable of incorporating global image features to enhance keypoint identification. This is particularly beneficial for distinguishing mirror-symmetric keypoints, such as the left and right eyes, which can be highly ambiguous when relying solely on local image features. Our ablation study demonstrates the performance improvements achieved through the utilization of this component.

## 3.4 Multimodal LLM for Keypoint Comprehension

Given a query image and an optional prompt specifying the keypoint of interest, our goal is to generate textual descriptions and keypoint locations that convey fine-grained keypoint information within the image. Recognizing the exceptional ability of LLMs in handling multimodal tokens for different perception tasks [76, 10, 77, 73, 78], we further leverage LLM for keypoint comprehension, which could effectively process various inputs: (1) the visual tokens $\mathbf{z}_q$ of the query image, (2) the prompt tokens $\mathbf{z}_p$, and (3) a sequence of language tokens $\mathbf{t}$, which depend on the three semantic keypoint comprehension scenarios.

**Keypoint Semantic Decoding.** We design the model to directly generate textual descriptions that interpret keypoint semantics, following the standard approach used by LLMs for text generation. Generally, the architecture of an LLM typically comprises Transformer layers (`TransformerLayers`) followed by a final Feed Forward Network (`FFN`). The latent embedding $\mathbf{u}$, which captures the fused multimodal information, can be computed as:

$$\mathbf{u} = \texttt{TransformerLayers}([\mathbf{z}_q, \mathbf{z}_p, \mathbf{t}]), \tag{3}$$

where $[\mathbf{z}_q, \mathbf{z}_p, \mathbf{t}]$ denotes the concatenation of the visual, prompt, and language tokens. This embedding $\mathbf{u}$ is then passed through the `FFN` and a `Softmax` function to generate the probability distribution $\mathbf{p}$ over the vocabulary for the next token:

$$\mathbf{p} = \texttt{Softmax}(\texttt{FFN}(\mathbf{u})). \tag{4}$$

**Keypoint Position Decoding.** Inspired by [78, 73], we introduce a special token, `<keypoint>`, into the vocabulary. Consequently, the 2D keypoint position $\mathbf{y}$ can be computed from the output latent embedding $\mathbf{u}_{\text{kpt}}$ of the special token using another `FFN` prediction head:

$$\mathbf{y} = \texttt{FFN}(\mathbf{u}_{\text{kpt}}) \in \mathbb{R}^2. \tag{5}$$

**Indentify-then-Detect (ItD) Strategy**. Instead of relying on extensive data fitting to learn fixed keypoint localization patterns, we adopt an approach where the LLM initially identifies the semantics of keypoints. Subsequently, a chain-of-thought process is employed to accurately detect the locations of these keypoints. Experiments demonstrate improved performance compared to alternative baselines.

### 3.5 Training and Inference Details

To retain the learned general knowledge of the pre-trained LLM, we employ LoRA [79] for efficient fine-tuning of LLM, while fully fine-tuning other modules of the framework. The training and inference processes for different tasks are outlined below.

**Keypoint Semantic Understanding.** As shown in Fig. 1-(a), this task focuses on extracting semantic textual information associated with specific keypoints within an image. The training objective is to minimize the language modeling loss, computed as the cross-entropy loss over the vocabulary of the LLM's tokenizer. Specifically, the loss function is defined as:

$$\mathcal{L} = \mathcal{L}_{\text{lm}}(\mathbf{a}, \hat{\mathbf{a}}), \tag{6}$$

where $\mathbf{a}$ is the text response predicted by the model, and $\hat{\mathbf{a}}$ is the ground-truth text response. During inference, given an image provided by the user and the corresponding keypoint position as a prompt, our model comprehends and generates the semantic meaning of the specified keypoint.

**Visual Prompt-based Keypoint Detection.** This task involves simultaneously comprehending the semantics of keypoints, generating textual descriptions of this understanding, and precisely localizing the keypoint coordinates, as shown in Fig. 1-(b). The overall training function further incorporates the L1 loss for keypoint regression:

$$\mathcal{L} = \lambda \|\mathbf{y} - \hat{\mathbf{y}}\| + \mathcal{L}_{\text{lm}}(\mathbf{a}, \hat{\mathbf{a}}), \tag{7}$$

where $\hat{\mathbf{y}}$ is the ground-truth keypoint position, and $\lambda$ is the loss weight that balances the learning of keypoint regression and text generation ($\lambda = 2$ in our implementation). During inference, the user provides two images: one as the query image for testing and the other as the support image for reference. Additionally, the keypoint definition for the support image should be provided as the support keypoint prompt. Our model then comprehends the semantics of the desired keypoint and detects its corresponding position in the query image.

**Textual Prompt-based Keypoint Detection.** As illustrated in Fig. 1-(c), this task aims to accurately localize keypoints based on the detailed keypoint descriptions. The loss function aligns with that of the visual prompt-based keypoint detection. During inference, users have the option to provide detailed descriptions of the desired keypoints or simply the keypoint names, based on which our model can detect the corresponding keypoints.

## 4 Experiments

### 4.1 Experimental Setup

#### 4.1.1 Datasets

In our experiments, we employ two datasets to evaluate the semantic keypoint comprehension in three scenarios: (1) The MP-100 dataset [21] for both *Keypoint Semantic Undertanding* and *Visual Prompt-based Keypoint Detection*: This dataset is a pioneering dataset for category-agnostic pose estimation, which encompasses 100 different object categories with over 20,000 instances. The number of keypoints varies across categories, ranging from 8 to 68. Following the protocols established by POMNet [21], the dataset is divided into five distinct splits to ensure comprehensive coverage across different model training and validation scenarios. Each split contains all 100 categories, with 70 for training, 10 for validation, and 20 for testing. The splits are carefully designed to avoid category overlap, maintaining the independence and integrity of training and testing scenarios.

(2) The AP-10K dataset [32] for *Textual Prompt-based Keypoint Detection*: The dataset comprises 23 animal families and 54 species, totaling 10,015 images. Each image is annotated with 17 keypoints, including two eyes, one nose, one neck, two shoulders, two elbows, two knees, two hips, four paws, and one tail. We follow CLAMP [23] to assess the models' ability to generalize to previously unseen animal species within a zero-shot learning paradigm. We establish two experimental scenarios based on the taxonomic relationship between the species in the training and test sets—specifically, whether they belong to the same animal order. Species within the same order typically share similar visual characteristics, whereas those from different orders exhibit greater diversity in appearance. These scenarios enable us to assess how different methods perform when generalizing to unseen species under varying conditions. Following CLAMP, we assign Bovidae and Canidae as the training and

Table 2: **Visual Prompt-based Keypoint Detection** on MP-100 [21] dataset. Performance (PCK) under 1-shot and 5-shot settings.

| Methods | 1-shot | | | | | | 5-shot | | | | | |
|---|---|---|---|---|---|---|---|---|---|---|---|---|
| | Split1 | Split2 | Split3 | Split4 | Split5 | Mean | Split1 | Split2 | Split3 | Split4 | Split5 | Mean |
| ProtoNet [80] | 46.05 | 40.84 | 49.13 | 43.34 | 44.54 | 44.78 | 60.31 | 53.51 | 61.92 | 58.44 | 58.61 | 58.56 |
| MAML [81] | 68.14 | 54.72 | 64.19 | 63.24 | 57.20 | 61.50 | 70.03 | 55.98 | 63.21 | 64.79 | 58.47 | 62.50 |
| Finetune [82] | 70.60 | 57.04 | 66.06 | 65.00 | 59.20 | 63.58 | 71.67 | 57.84 | 66.76 | 66.53 | 60.24 | 64.61 |
| POMNet [21] | 84.23 | 78.25 | 78.17 | 78.68 | 79.17 | 79.70 | 84.72 | 79.61 | 78.00 | 80.38 | 80.85 | 80.71 |
| CapeFormer [22] | 89.45 | 84.88 | 83.59 | 83.53 | 85.09 | 85.31 | 91.94 | 88.92 | 89.40 | 88.01 | 88.25 | 89.30 |
| KptLLM | **91.66** | **86.58** | **86.19** | **84.76** | **86.32** | **87.10** | **93.17** | **89.45** | **90.08** | **88.74** | **89.52** | **90.19** |

test sets for the different order setting, while Canidae and Felidae are chosen as the training and test sets for the same order setting.

### 4.1.2  Evaluation & Metrics

Different evaluation methods and metrics are used for different tasks of semantic keypoint comprehension. (1) *Keypoint Semantic Understanding*: We use the MP-100 dataset [21] (Split-1), with the keypoint semantic labels adopted from X-Pose [52]. Some keypoints are excluded from the evaluation due to ambiguity or inadequacy in their descriptions, such as those used to describe the collar in the clothing category. By aggregating the results of tested keypoints, we derive corresponding accuracy rates (%). (2) *Visual Prompt-based Keypoint Detection*: We employ the Probability of Correct Keypoint (PCK) metric, which is the standard evaluation measure for this task. Consistent with POMNet [21], we uniformly set the PCK threshold to 0.2 across all categories. Additionally, we compute and report the average PCK over all five data splits to provide a comprehensive indication of the model's overall effectiveness. (3) *Textual Prompt-based Keypoint Detection*: Following CLAMP, we employ average precision (AP) as the primary metric for AP-10K. This metric is computed based on the object keypoint similarity (OKS). For detailed protocol definitions, please refer to [24].

### 4.1.3  Implementation Details

**Architecture.** We utilize LLaVA-V1.5-7B [10] as our base model, which incorporates the ViT-based visual encoder of CLIP for image encoding and Vicuna-7B (fine-tuned from Llama-2) as the LLM backbone. We employ LoRA for efficient fine-tuning LLM. Instead, all other modules, including the visual encoder, prompt encoder, prompt feature extractor, and a series of linear layers and feed forward networks, undergo full fine-tuning. The input image only contains a single object of interest, cropped according to the ground-truth bounding box and resized to 336×336, consistent with CLIP-ViT-L.

**Training Details.** LoRA parameters are configured with a rank of 128 and an alpha of 256. Optimization is conducted using AdamW, with a learning rate of 2e−4 and weight decay of 0. We utilize 8 NVIDIA A100-80G GPUs for training, and use the DeepSpeed engine to enhance training efficiency. Each GPU operates with a batch size of 16, and we employ a gradient accumulation step of 1.

### 4.2  Keypoint Semantic Understanding

As depicted in Tab. 1, we present the accuracy for keypoint semantic understanding on MP-100 [21] Split-1 set. To facilitate a comprehensive comparison, we highlight the keypoint area in the image and feed the processed image, along with the task instruction, into LLaVA [10]. We report the performance of both the original LLaVA model and a version fine-tuned on the MP-100 dataset. The original LLaVA performs notably poorly in grasping keypoint semantics, indicating the

Table 1: **Keypoint Semantic Understanding** on MP-100 (Split-1) [21].* means LLaVA is finetuned using LoRA.

| Methods | Accuracy |
|---|---|
| LLaVA [10] | 3% |
| LLaVA∗ [10] | 72% |
| KptLLM | **83%** |

inadequacy of traditional multimodal large language models in capturing fine-grained semantic details. Conversely, the fine-tuned LLaVA demonstrates significantly enhanced performance, thereby validating the efficacy of our training pipeline. Furthermore, our KptLLM surpasses the fine-tuned LLaVA by a substantial margin, particularly in terms of keypoint accuracy (83% vs 72%). It demonstrates the effectiveness of our keypoint prompt token in guiding attention to the fine-grained keypoint area.

Table 3: **Visual Prompt-based Keypoint Detection** for cross super-category evaluation on MP-100 [21]. Experiments are conducted under the 1-shot setting.

| Method | Human Body | Human Face | Vehicle | Furniture |
|---|---|---|---|---|
| ProtoNet [80] | 37.61 | 57.80 | 28.35 | 42.64 |
| MAML [81] | 51.93 | 25.72 | 17.68 | 20.09 |
| Fine-tune [82] | 52.11 | 25.53 | 17.46 | 20.76 |
| POMNet [21] | 73.82 | 79.63 | 34.92 | 47.27 |
| CapeFormer [22] | 83.44 | 80.96 | 45.40 | 52.49 |
| KptLLM | **83.91** | **83.37** | **46.23** | **54.05** |

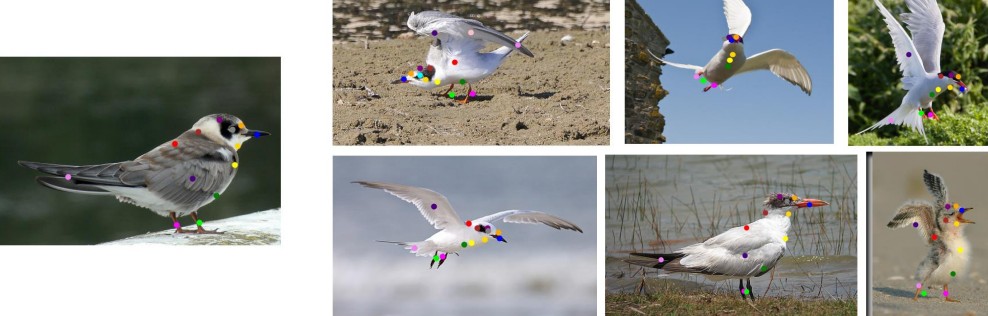

(a) Support Image with Keypoints    (b) Query Image with Model Predictions

Figure 3: Using the same support image with support keypoints, our model could effectively detect different query images with various poses, object appearances, and environments.

### 4.3 Visual Prompt-based Keypoint Detection

**1-shot & 5-shot Evaluation**. We compare our method with the previous visual prompt-based methods ProtoNet [80], MAML [81], Fine-tune [82], POMNet [21], and CapeFormer [22]. Tab. 2 presents the PCK results of different approaches on the MP-100 dataset under both 1-shot and 5-shot settings. Compared with previous methods, KptLLM showcases the potential of MLLM in detecting keypoints through the use of visual prompts, consistently outperforming across all settings and data splits. More importantly, we integrate keypoint semantic understanding into the output response, introducing novel functionalities for comprehending the semantic aspects of support image keypoints.

**Cross Super Category Evaluation.** To thoroughly assess generalization across markedly different categories, we conduct a cross-supercategory evaluation following the protocol of POMNet [21]. While the MP-100 dataset ensures that training, validation, and test categories are non-overlapping, some categories may still exhibit similar features, *e.g.*, body characteristics commonly shared among different quadruped animals. To address this, we designate four supercategories—human face, human body, vehicle, and furniture—from the MP-100 dataset as test categories. The remaining categories are utilized for training, allowing us to better evaluate the model's ability to generalize across significantly diverse categories. As shown in Tab. 3, KptLLM consistently outperforms previous methods, highlighting the robustness and excellent generalization ability of our proposed method.

**Qualitative Results.** For the visual prompt-based keypoint detection task, the input necessitates a support image of the object to be tested, as well as keypoint positions that represent the definitions of those keypoints. In this study, we examine how the visual disparity between support images and query images affects the model's performance. As depicted in Fig. 3, our model is capable of effectively detecting keypoints in various query images when provided with the same support image and its corresponding keypoints. This effectiveness is maintained even in the presence of differences in object poses, appearances, and environmental conditions.

Table 4: **Textual Prompt-based Keypoint Detection** on AP-10K [32].

| Method | Train | Test | $AP$ | $AP_{50}$ | $AP_{75}$ | $AP_M$ | $AP_L$ | $AR$ |
|---|---|---|---|---|---|---|---|---|
| SimpleBaseline [48] | Bovidae | Canidae | 41.3 | 79.4 | 36.4 | 26.8 | 41.3 | 49.1 |
| CLAMP [23] | Bovidae | Canidae | 46.9 | 84.4 | 45.6 | **30.3** | 46.9 | 53.8 |
| KptLLM | Bovidae | Canidae | **62.2** | **94.5** | **68.7** | 26.0 | **62.3** | **65.8** |
| SimpleBaseline [48] | Canidae | Felidae | 39.6 | 74.1 | 34.5 | 9.5 | 40.2 | 46.6 |
| CLAMP [23] | Canidae | Felidae | 48.4 | 85.7 | 44.0 | 13.6 | 48.9 | 55.1 |
| KptLLM | Canidae | Felidae | **70.2** | **97.8** | **82.2** | **49.2** | **70.6** | **74.7** |

Table 5: Ablation study of semantic understanding.

| Strategies | PCK |
|---|---|
| w/o ItD | 87.68 |
| w/ ItD | **91.66** |

Table 6: Ablation study of prompt feature extraction.

| Strategies | PCK |
|---|---|
| Average Pooling | 89.78 |
| Ours | **91.66** |

Table 7: The effect on combining visual and textual prompts.

| Visual | Textual | PCK |
|---|---|---|
| ✓ | | 91.66 |
| ✓ | ✓ | **92.18** |

## 4.4 Textual Prompt-based Keypoint Detection

The results are presented in Tab. 4. Compared to previous textual prompt-based model-CLAMP [23], KptLLM achieves superior cross-species generalization. Specifically, our model demonstrates a 15.3 average precision (AP) improvement in the different order setting and a 21.8 AP increase in the same order setting. Notably, our model performs better in the same order setting, where species often share similar visual characteristics. Overall, we show that leveraging detailed keypoint descriptions through comprehensive text, combined with commonsense knowledge from LLMs, effectively enhances generalizable performance in keypoint localization.

## 4.5 Ablation Study

In this subsection, we perform ablation study on the design choices of our model. The experiments are conducted on the visual prompt-based keypoint detection task using the MP-100 Split-1 setting with the PCK metric reported.

**Indentify-then-Detect (ItD) Strategy.** KptLLM follows the identify-then-detect paradigm, where the model learns to first interpret the semantic information of the keypoint to be detected, and then predict the precise location of the keypoint. In Tab. 5, we validate the effectiveness of our ItD strategy. We observe notable enhancement, which can be attributed to the inter-task synergy that arises from the ItD mechanism.

**Prompt Feature Extractor.** In Tab. 6, we compare our prompt feature extractor (Sec. 3.3) with the average pooling based feature extraction method [21]. The results show that our prompt feature extractor significantly outperforms the baseline method (91.66 vs 89.78), which validates the efficacy of our prompt feature extractor in enhancing focus on fine-grained keypoint areas.

**Combining Visual and Textual Prompts.** In Tab. 7, rather than relying solely on the visual prompt for localization, we further incorporate the textual prompt to demonstrate the effect of this combination. The improved results indicate that the textual prompt could provide valuable high-level and semantically rich guidance, enhancing keypoint localization.

## 5 Conclusion

This paper introduces the novel challenge of *Semantic Keypoint Comprehension*, which aims to comprehend keypoints across different task scenarios. To address this challenge, we present KptLLM, a novel and unified multimodal large language model designed to adeptly process various modality inputs, facilitating the interpretation of both semantic contents and keypoint locations. Extensive experiments show the superiority of our model in three different tasks for comprehending keypoints, including keypoint semantic understanding, visual prompt-based keypoint detection, and textual prompt-based keypoint detection. We hope this work can open up new possibilities for more fine-grained multimodal vision-language understanding and provide valuable insights for future research.

# 6 Discussion

**Limitations.** (1) A major limitation of our work is the model's size and computational efficiency, which is a common challenge for MLLMs compared to traditional vision models. However, this is acceptable because, as a pioneering effort in utilizing LLMs for keypoint comprehension, our main contribution is demonstrating the potential of LLMs to understand and locate pixel-level details at keypoints. (2) Additionally, the datasets used in our experiments have constraints in the diversity of object and keypoint categories for both training and testing. This highlights the need to expand these datasets to validate the model's applicability in more diverse, real-world scenarios.

**Future Work. (1) Improving the Capacity of the Vision Encoder:** Our work follows LLaVA [10] by employing a CLIP-based ViT as the vision encoder. However, some studies [83, 84] have demonstrated that stronger vision encoders can lead to more significant improvements, e.g., DINOv2 [85]. **(2) Refining Keypoint Decoding Strategy:** Inspired by previous MLLMs for perception tasks [78, 73], we introduce a special token `<keypoint>` into the model's vocabulary. When the model generates this `<keypoint>` token, its hidden embedding is decoded to the corresponding keypoint position. Although this strategy has shown promising results, it remains sub-optimal for user interaction. A more direct approach is to output the keypoint coordinates as textual descriptions. However, training a model to express numerical values in text using cross-entropy loss is challenging because slight deviations in numerical values can lead to significant differences in the generated text. Therefore, it intuitively requires more data for effective training. **(3) Expanding Data Scale and Category Diversity:** The datasets used in our experiments follow standard benchmarks. However, both the MP-100 dataset [21] for visual prompt-based keypoint detection and the AP-10K dataset [32] for textual prompt-based keypoint detection contain only a small amount of data, which limits the model's generalization performance. Furthermore, the limited diversity of object and keypoint categories greatly reduces the model's applicability, making it insufficient for handling open-world scenarios. A promising direction is to leverage large-scale keypoint datasets for training, such as UniKPT [52], which could further explore the upper bounds of MLLMs for keypoint comprehension.

**Broader Impact.** The study aims to enhance MLLMs for understanding images at a more granular keypoint level. We also propose a new challenge of keypoint semantic understanding, which holds promise for benefiting tasks such as structural understanding, action recognition, and medical image analysis. Nevertheless, recognizing the potential negative impacts that are common to many MLLMs, our model also carries risks, including the amplification of societal biases and concerns regarding privacy and ethics. To address these issues, we are committed to implementing safeguards, including strict access controls and the establishment of clear usage policies and agreements.

## Acknowledgements

The work is partially supported by the Young Scientists Fund of the National Natural Science Foundation of China under grant No.62106154, by the Natural Science Foundation of Guangdong Province, China (General Program) under grant No.2022A1515011524, and by Shenzhen Science and Technology Program JCYJ20220818103001002, and by the Guangdong Provincial Key Laboratory of Big Data Computing, The Chinese University of Hong Kong (Shenzhen).

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
