# OpenReview forum: "KptLLM: Unveiling the Power of Large Language Model for Keypoint Comprehension"
_NeurIPS.cc/2024/Conference — NeurIPS 2024 poster_

### Official Review · Reviewer_7RyA · 2024-07-03

**Soundness:** 3
**Presentation:** 3
**Contribution:** 3
**Rating:** 7
**Confidence:** 3

**Summary:**

The paper introduces and studies the problem of Semantic Keypoint Comprehension to evaluate the capability of Multimodal Large Language Models (MLLMs) in tackling fine-grained perception and comprehension. It does so by defining three related tasks: (1) keypoint semantic understanding, (2) Visual prompt-based keypoint detection and  (3) Textual Prompt-based Keypoint Detection. The first task targets the semantic interpretation capabilities of objects and keypoints, while the other two tasks involve position detection and semantic comprehension from visual or textual descriptions. In the context of this paper, keypoint detection (a.k.a object pose estimation) is the task of localizing the semantic keypoints of objects.

The authors further present KptLLM, an MLLM for Semantic Keypoint Comprehension based on LLaVA. The model employs an identify-then-detect strategy, where the model first captures the required semantics and then detects the precise keypoint position. The model employs attention to fuse features from image and keypoint encoded prompts. The fused encoding and the encoding of visual inputs are used as additional latent contexts for an LLM, which is finetuned with LORA to generate the prompt. The keypoint position is further regressed with an MLP head. The authors evaluate their proposed model on the MP-100 (diverse and category agnostic) dataset and the AP-10K dataset and compare its performance to state-of-the-art methods for keypoint detection and to pretrained and finetuned LLaVA models.

**Strengths:**

- The paper introduces new tasks for studying the semantic comprehension of MLLMs at a fine-grained level
- The paper presents an MLLM-based method for tackling the introduced tasks, which outperforms current keypoint detection methods across datasets and tasks by notable margins
- The proposed method also outperforms state-of-the-art MLLM (pre trained and finetuned) on the target tasks
- The authors ablate their proposed detection strategy and other architectural choices.

**Weaknesses:**

- The properties (and potential limitations) of the MLLM-based approaches in terms of runtime, memory, and scalability to multiple key points, compared to existing state-of-the-art methods, are not discussed
- Lack of details on the methods chosen for cross-comparison and their key properties/assumptions makes it harder to evaluate the contribution
While the paper is overall well structured, the evaluation protocol and details are not always easy to follow

**Questions:**

- Can you please provide more details on runtime, memory, and scalability?
- Is the visual encoder trained/fine-tuned or frozen? The textual description mentions a pretrained frozen encoder, while the image shows it to be fine-tuned

**Limitations:**

It is worth elaborating on memory, runtime and scalability aspects of the proposed method.

---

> ### Author Rebuttal · Authors · 2024-08-07
>
> ### Q1: The runtime, memory, and scalability of our model
>
> During inference, our model processes multiple keypoints by stacking them into a batch. For the standard 17 keypoints, our model obtains their positions in 3.45 seconds, consuming 25,100 MiB of memory. It is important to note that in our implementation, we did not utilize LLM-related acceleration and optimization techniques, which could potentially enhance the model's efficiency.
>
> ### Q2: Details of comparable methods and evaluation protocol
>
> **Details of comparable methods:** For visual prompt-based keypoint detection, all the methods take the visual prompt and query image as the inputs, and output the corresponding keypoints. These methods only rely on matching support keypoint features with query image features. This matching is semantic-agnostic, which can fail when there is insufficient similarity between the support and query instances, especially when they differ significantly in poses, textures, or styles. Additionally, these methods struggle when dealing with similar keypoints that exhibit symmetrical appearances, such as the left eye and right eye. For textual prompt-based keypoint detection, all the methods take the textual prompt and test image as the inputs, and output the corresponding keypoints.
>
> **Details of evaluation protocol:** For **keypoint semantic understanding** and **visual prompt-based keypoint detection**, we employ the standard benchmark-MP100 for evaluation. This benchmark ensures that the species categories used in the training and testing phases are non-overlapping, meaning the test objects are entirely unseen during training. Following the standard evaluation protocol used by comparable methods, we sample 3,000 random pairs (one as the support image and another as the query image) for each novel category during testing. With 20 test categories for each split, we construct a total of 60,000 pairs for evaluation. For **textual prompt-based keypoint detection**, we follow CLAMP to evaluate the models’ generalization ability on unseen animal species in the zero-shot learning paradigm. Two experimental settings are defined based on whether the animal species in the training set and test set belong to the same animal order or not. Species belonging to the same order tend to have similar appearances, while species from different orders exhibit more diverse appearances.
>
> ### Q3: About Visual Encoder
>
> We have finetuned the visual encoder, as the original pre-trained CLIP image encoder cannot adequately capture the pixel-level fine-grained visual information, which is essential for the keypoint detection tasks.

---

> > ### Comment · Reviewer_7RyA · 2024-08-12
> >
> > Thank you for addressing my questions. Overall, the soundness and presentation of the presented work are good: the paper describes an interesting problem and a method for tackling it while demonstrating superior results and performing ablations, so I keep my original rating for these criteria. However, after reading through the comments and concerns made by the other reviewers and the authors' responses regarding the contribution of the task presented, I have decided to change my original rating from excellent to good.

---

### Official Review · Reviewer_wNmk · 2024-07-10

**Soundness:** 3
**Presentation:** 3
**Contribution:** 3
**Rating:** 6
**Confidence:** 5

**Summary:**

This paper proposes an LLM-based keypoint localization method that can detect keypoint by visual or text prompt and classify the keypoint category by given its coordinates. Experiments on several public benchmarks demonstrate the effectiveness of the proposed method.

**Strengths:**

Adopting LLM to perform in-context learning of keypoint localization is interesting and this paper demonstrates that this pipeline is feasible and can get superior performance.

**Weaknesses:**

1.	It seems that the Visual Prompt-based Keypoint Detection and Textual Prompt-based Keypoint Detection are trained separately. However, it is more useful if we can complete these tasks using a single model. So how about jointly training above two tasks? Can they benefit from each other or not?
2.	What’s the meaning of keypoint identifiers in Table 2 and 3? It would be confusing if these is no specific illustration.
3.	Some typos. “Texual”->Textual in line217, 234, 254, etc.
4.	Some claims are not proper. For instance, in line 57-58 “To the best of our knowledge, this is the first work that equip LLMs with the point-level visual perception capabilities.” [A] already introduces LLM into keypoint localization and this paper should emphasize the different aspect such as  visual prompt-based keypoint localization.

[A] D. Wang, S. Xuan, S. Zhang. LocLLM: Exploiting Generalizable Human Keypoint Localization via Large Language Model. CVPR 2024.

**Questions:**

In Weakness.

**Limitations:**

This paper discusses the limitation in A.3.

---

> ### Author Rebuttal · Authors · 2024-08-07
>
> ### Q1: The mutual benefit between textual prompt and visual prompt
>
> In the main article, since textual prompt-based and visual prompt-based keypoint detection have their own benchmarks, combining both prompts for training would result in unfair comparisons. Here, we supplement the experiment by using both textual and visual prompts. The results on MP100 split-1 show that the textual prompt could provide valuable high-level and rich semantic guidance to enhance the keypoint localization.
>
> | Visual  Prompt    | Textual  Prompt   | MP100-Split1 |
> |------------|------------|--------------|
> | &#10004; |            | 91.66        |
> | &#10004; | &#10004; | 92.18        |
>
> Table: The ablation study of the mutual benefit between textual prompt and visual prompt
>
>
> ### Q2: The explanation of keypoint identifier
>
> The keypoint identifier used in the Capeformer can alleviate the ambiguity of keypoints with similar appearances when compared to using only visual features for low-level matching. However, the keypoint identifier requires additional effort to annotate keypoints across different categories, which limits real-world applications where keypoint identifiers are not available.
>
>
> ### Q3: About typo
>
> Thanks for pointing out the typo errors. We will correct them in the revised version.
>
>
> ### Q4: About related work
>
> LocLLM [A] was publicly available on June 7, 2024, which is later than our submission date. Therefore, our work can be considered concurrent. LocLLM only uses textual prompts for human keypoint localization using LLM. In contrast, we explore the use of LLM for more comprehensive keypoint comprehension of diverse objects. We will revise some claims and add discussions to address this.

---

### Official Review · Reviewer_Hwow · 2024-07-12

**Soundness:** 2
**Presentation:** 3
**Contribution:** 2
**Rating:** 6
**Confidence:** 3

**Summary:**

The authors aim to enhance multi-modal LLMs with semantic keypoint comprehension. They introduce a hybrid visual prompting approach using a query and a support input image. In their pipeline, visual features from both images are extracted via a vision encoder model. Additionally, a support keypoint prompt, indicating keypoint positions on the support image, is used. The embeddings from this prompt are combined with the support image embeddings and fed into the LLM. This mixed embedding helps the model detect and understand the keypoint in the input query image, generating the desired keypoint location information.

**Strengths:**

The paper is well written and easy to follow.

The proposed method shows promising results albeit its scope of usability and also generalization should be better examined.

**Weaknesses:**

The proposed support image and support keypoint prompt essentially act as few-shot examples for the model. It would be beneficial to explicitly acknowledge this in the paper.

Another limitation of the proposed method is the requirement for a visually similar support image with keypoint locations for each input query image. Have the authors tested the model with support images that are visually different from the query images? How much visual difference can the model handle without a significant drop in accuracy? Conducting an ablation study on this would be crucial to determine the extent of semantic (high-level) or texture (low-level) distribution shifts the method can tolerate without a notable accuracy decline.

**Questions:**

The authors should compare their approach to GPT-4V as a baseline to better understand the benefits of the additional steps they have added to their proposed pipeline.

It is unclear where the support image and support keypoint prompts come from during inference. Are these data available to other methods in the experiments?

Figure 2 is somewhat confusing. (Z_q) is fed to the LoRA section, while the rest of the information is fed to the rest of the LLM. Is the figure designed this way on purpose? If so, could you please clarify this?

**Limitations:**

The authors have discussed the limitations of their work.

---

> ### Author Rebuttal · Authors · 2024-08-07
>
> ### Q1: Visual prompt-based keypoint detection is equal to few-shot keypoint detection
>
> The visual prompt-based keypoint detection can be considered as the few-shot keypoint detection.
> In our main article, we have explicitly illustrated the input requirements of visual prompt-based keypoint detection.
>
>
>
> ### Q2: The impact of visual differences between support images and query images on model performance
>
> In fact, visual prompt-based keypoint detection does not require a visually similar support image with keypoint locations for each input query image.
> In the MP-100 dataset, objects within the same novel category are not always similar, and there are many challenging cases. To illustrate this, we select the tern body as a novel category, which exhibits significant species variation, to demonstrate the impact of visual differences between support images and query images on model performance. As shown in the visualizations in the PDF, using the same support images with support keypoints, our model can effectively detect different query images, despite variations in poses, object appearances, and environments.
>
> ### Q3: Compared with GPT-4V
>
> We compare our model with GPT-4V for visual prompt-based keypoint detection. Given the high cost of utilizing the GPT-4V API, we randomly sample 10 paired images for each category in the test set. For the split-1 set, which includes 20 test categories, this results in a total of 200 image pairs used for evaluation. As shown in the table below, our model effectively detects novel objects given the support image and support keypoints, outperforming GPT-4V in these scenarios.
>
> | Method | MP100-Split1-Sub |
> |--------|------------------|
> | GPT-4V | 25.23            |
> | Ours   | 91.58            |
>
> Table: Compared with GPT-4V for visual prompt-based keypoint detection.
>
> ### Q4: The evaluation protocol is consistent across our model and other comparable methods
>
>
> Our model and other comparable methods all use the standard benchmark MP-100 for the evaluation of visual prompt-based keypoint detection task. This benchmark ensures that the species categories used in the training and testing phases are non-overlapping, meaning that the test objects are entirely unseen during training. Following the standard evaluation protocol used by comparable methods, we sample 3,000 random pairs (one as the support image and another as the query image) for each novel category during testing. With 20 test categories for each split, we construct a total of 60,000 pairs for comprehensive and effective evaluation.
>
>
> ### Q5: About the figure of our framework
>
> We apologize for any misunderstanding.
> LoRA (Low-Rank Adaptation) is an efficient fine-tuning technique that modifies existing layers of a Large Language Model (LLM) by introducing additional low-rank matrices. These matrices enhance the model's ability to comprehend keypoints without the need for extensive retraining of the original weights. As a result, all multimodal inputs ($Z_q$ and other inputs) are processed through both the LLM and the LoRA weights.

---

> > ### Comment · Reviewer_Hwow · 2024-08-12
> >
> > I appreciate the authors’ rebuttal.
> >
> > Q5: I did not ask “What is LoRA,” which I already know. I asked why the figure is designed in that particular way; the figure is confusing for the average reader. However, this question and the answer do not affect my rating.
> >
> > Considering the other reviewers’ feedback and the authors’ rebuttal, I plan to increase my overall rating. Before doing so, I would like to know if the authors plan to include the reviewers’ feedback in their final camera-ready version (specifically for my reviews, Q2 and Q3), as I did not see any promises being made by the authors in their comments.

---

> > > ### Author Response · Authors · 2024-08-12
> > > **Rebuttal by Authors**
> > >
> > > Thank you for your response. We greatly appreciate all the valuable feedback from the reviewers, which will help us improve our paper. We promise that all reviewers' feedback, including yours, will be incorporated into our final camera-ready version. Specifically, your comments, along with the visualizations and additional experiments in response to Q2 and Q3, will further clarify the advantages of our method for visual prompt-based keypoint detection. Additionally, we appreciate your suggestions regarding the framework diagram, and we will revise it to prevent any misunderstandings.

---

> > > > ### Comment · Reviewer_Hwow · 2024-08-12
> > > >
> > > > Considering the additional details and clarifications provided by the authors in their rebuttal, along with their commitment to incorporate the rebuttal and reviewers’ comments in the camera-ready version, I believe the manuscript meets the acceptance criteria for a poster presentation. Therefore, I increase my rating to 6.

---

### Official Review · Reviewer_j2Jb · 2024-07-13

**Soundness:** 3
**Presentation:** 3
**Contribution:** 2
**Rating:** 4
**Confidence:** 4

**Summary:**

The paper proposes to build a MLLM for keypoint detection. They propose three main points to achieve this. First, the paper introduces a new benchmark that measures various types of keypoint detection including both text and visual-prompt-based, as well as traditional keypoint understanding. Next, they propose to build a multimodal LLM, which is designed specifically for keypoint detection that is shown to perform quite well on this task.

**Strengths:**

[S1] Good results: The paper boasts quite good results for the task at hand. The experiments are thorough and clearly outline the contribution of various components of the model.

[S2] New benchmark: The paper proposes a new benchmark for keypoint understanding, which goes beyond the traditionally available keypoint understanding benchmarks by bringing visual and text prompting. While this is mainly in service to the current paper, it could be independently useful for posterity.

**Weaknesses:**

[W1] Not a lot of novel insights: This is my main, and mostly only, weakness for the paper, but it take a few different forms. Firstly, this is a straightforward "building" of MLLM formula; i.e., convert the task at hand into "instruction format" -> train a vision to LLM connector --> fine-tune LLM; that is used by a lot of recent works. As such, this would be mostly of interest to the domain performance, Keypoint comprehension in this case. This already puts some doubt as to whether this is of interest enough to a broad community. Next, there are also not a lot of modular components that might be broadly useful that hasn't already been studied by other works. Some details below:

W1.1 LLava does almost as good (for supported tasks): The paper shows that, once fine-tuned, the popular LLAVA model does almost as good as the proposed method. The significance of this finding is not thoroughly discussed. Why build a new model at all? Furthermore, there are more capable MLLMs baseline such as R1, R2, R3 that might be more suitable targets for fine-tuning. See 1.2 below as well

W1.2 The proposed method of generating a special token that can be decoded to a key point is also explored before (R1). Besides, there are also various MLLMs that are capable of doing region comprehension (R2, R3...). Those are not sufficiently discussed. If fine-tuned, like LLAVA, how would they do on this task?

W1.3 Is visual prompting a good paradigm for keypoint detection? What is the importance of visual prompting for this task? [EDIT: Clarified in post-rebuttal]

References:
R1: https://arxiv.org/abs/2311.06612
R2: https://arxiv.org/abs/2306.15195
R3: https://openreview.net/forum?id=2msbbX3ydD

**Questions:**

Please address the weaknesses above, which also contain questions.

---

> ### Author Rebuttal · Authors · 2024-08-07
>
> ### Q1: Novelty and Contribution
>
> **To Keypoint Detection Community:** To the best of our knowledge, this is the first work to
> address the problem of semantic keypoint comprehension, which aims
> to understand keypoints within different human-AI interaction contexts.
> Previous keypoint detectors primarily rely on directly learning visual or textual prompts for keypoint localization through extensive data fitting, often neglecting the semantic understanding of keypoints. This oversight often leads to misinterpretations of the prompts and inaccurate predictions. In contrast, our work proposes the "identify-then-detect'' strategy, which requires the model to first comprehend the semantics of keypoints and then accurately determine their positions through a chain-of-thought process. This paradigm could enhance both interpretability and localization accuracy.
>
>
> **To MLLM Community:**
> Compared with previous works (R1, R2, R3) that focus on region-level understanding, our model takes a further step to achieve more fine-grained, pixel-level understanding and localization, which is critical for various real-world applications. For instance, **semantic keypoint understanding** equips MLLMs with enhanced visual comprehension and analytical capabilities, essential for tasks such as structural analysis, action recognition, and medical image interpretation. **Visual prompt-based keypoint detection** allows MLLMs to acquire keypoint definitions from visual prompts, enabling cross-class and cross-keypoint localization using sample images provided by users. Additionally, **textual prompt-based keypoint detection** enables MLLMs to follow human language guidance, facilitating keypoint localization on arbitrary objects and keypoint categories in a zero-shot manner.
>
>
> ### Q2: Compared with LLaVA, R1, R2, R3
>
> The preliminary results in our paper indicate that LLaVA can address the keypoint semantic understanding task through fine-tuning, primarily because this task only involves answering questions based on images. Given the architecture of LLaVA and similar models like R1, R2, and R3, these models can be directly fine-tuned to support this task.
> **However, they all lack capabilities for fine-grained keypoint localization, a critical aspect that our article emphasizes.** For example, LLaVA, R1, R2, and R3 cannot support the input of visual prompts (support images along with support keypoint definitions) for keypoint detection, which could enhance the generalization capabilities of in-context learning within LLMs. In addition, although R1 employs a special token to extract features from LLMs and uses an additional decoder to decode corresponding masks or boxes, it does not address the new challenge of keypoint detection, which remains unhandled by existing models.
>
> ### Q3: The importance of visual prompt-based keypoint detection
> Actually,  visual prompt-based keypoint detection is a task that has been widely studied in keypoint detection community. It is also supported by a well-established benchmark, which further demonstrates its significance and practice.
> Moreover,  visual prompt-based keypoint detection is important and practical in various real-world scenerios. For example, real-world applications across various fields often require detecting keypoints on a variety of unseen objects.
> To address this, it is a good choice to use a visual prompt-based keypoint detection model, which can detect any poses of unseen objects. In such scenerio, compared to textual prompt-based keypoint detection, visual prompts could allow the model to more accurately detect keypoints that are difficult to describe semantically, such as cloth.

---

> > ### Comment · Reviewer_j2Jb · 2024-08-14
> > **Post rebuttal comments**
> >
> > Thank you for your detailed reply.
> >
> > > Response to Q1
> > I agree with some of the points and am unclear on others. Even the points that sounds good. E.g., "For instance, semantic keypoint understanding equips MLLMs with enhanced visual comprehension and analytical capabilities, essential for tasks such as structural analysis, action recognition, and medical image interpretation." is not shown in the paper. A stronger submission would indeed do a generic eval to highlight these properties as well.
> >
> > > Response to Q2
> > I kinda disagree that existing MLLMs cannot be fine-tuned for this task. However, I do not think that is a weakness by itself. MLLMs are very versatile and they could be fine-tuned to a broad range of tasks, that does not, by itself pose a weakness. However, I am worried about a slight overclaim in treating the current work as more than what it does, i.e., enabling a new decoding to take place similar to perception GPT.
> >
> > > Response to Q3
> > Thank you for your response. I understand that it is indeed an established benchmark and I was able to verify that. So, I stand corrected from my earlier comments.
> >
> > Overall, some of my concerns have been clarified but I think a major rewrite with additional experiments and discussion is needed. So, I will slightly raise my score (also in light of other reviews and responses) but I am still leaning towards a major revision is needed.

---

> > > ### Author Response · Authors · 2024-08-14
> > >
> > > Thanks for your response.
> > >
> > > - In addressing Q1, we emphasize the significance and potential applications of the proposed semantic keypoint detection task. However, these downstream applications are beyond the scope of our current work and are intended for future exploration. We will adjust some of our claims accordingly.
> > >
> > > - Regarding Q2, fine-tuning existing MLLMs for keypoint detection is a non-trivial task, as it involves challenges such as keypoint data preparation and necessary architectural modifications. This highlights the importance of our work, which pioneers the advancement of MLLMs for fine-grained keypoint understanding and localization. We will add some discussions to ensure clarity and avoid any potential misinterpretations.

---

### Author Rebuttal · Authors · 2024-08-07

We appreciate the efforts of all reviewers in reviewing our paper and providing insightful comments and valuable suggestions. The supplementary visualization results (To \# Reviewer Hwow) have been included in the rebuttal PDF.

---

### Author Response · Authors · 2024-08-12
**Official Comment by the Authors of Submission 9290**

Dear AC and Reviewers,

Thank you for taking the time to review our paper. We are delighted to see that the reviewers recognized the strengths of our work. Specifically, all the reviewers (j2Jb, Hwow, wNmk, 7RyA) acknowledged the great results we achieved. Moreover, Reviewers wNmk and 7RyA appreciated our contributions to advancing the semantic comprehension of MLLMs at a fine-grained level and the adoption of LLMs for in-context learning of keypoint localization.

However, the reviewers also raised some concerns. For instance, Reviewer j2Jb questioned the broader impact of our work, while Reviewer Hwow requested more detail on the visual prompt-based keypoint detection. We have provided a detailed rebuttal addressing these concerns and kindly ask the reviewers to consider our responses when convenient.

Thank you once again for your valuable feedback.

Sincerely,

The Authors of Submission 9290

---

### Decision · Program_Chairs · 2024-09-25

**Decision:**

Accept (poster)

**Comment:**

**Summary:** This work introduces KptLLM, an MLLM that addresses the challenges of semantic keypoint comprehension, where a model must reason about fine-grained semantic visual information through keypoint. KptLLM shows that using a support set and LoRA on a frozen LLM leads to improvements over other baseline models on keypoint comprehension.

**Strengths/Weaknesses:** The reviewers comment on the strength of the approach for an interesting problem and that KptLLM leads to noticeable improvements over baselines. Most weaknesses focus on a lack of clarity. The wording/figures/tables can be confusing and the authors do not explicitly say the approach using a support set is an instance of few-shot prompting. This can be addressed in the camera ready. Some other weaknesses (e.g. selecting less similar support sets to test generalization) were addressed in the rebuttal. The remaining comments from Reviewer j2Jb should be included more thoroughly in the camera ready as well.

**Recommendation:** I recommend this paper for acceptance. The reviews are positive overall and the authors appear to have thoughtfully addressed reviewers’ questions and concerns through additional experiments and examples. Reviewers have requested more clarity in the Figure 2 and Tables 5 and 6) As acknowledged during the rebuttal, I’d like to see all of the additional experiments and suggestions for language shifts included in the final version. A more in-depth analysis of Llava (expanding on the discussion with Reviewer j2Jb) should be included as well to further justify the contributions of KptLLM.